# Microbial Fuel Cell-Based Organic Matter Sensors: Principles, Structures and Applications

**DOI:** 10.3390/bioengineering10080886

**Published:** 2023-07-26

**Authors:** Huang Yao, Jialong Xiao, Xinhua Tang

**Affiliations:** School of Civil Engineering and Architecture, Wuhan University of Technology, Wuhan 430062, China

**Keywords:** MFC sensors, organic, BOD, COD, construction

## Abstract

Wastewater contains a significant quantity of organic matter, continuously causing environmental pollution. Timely and accurate detection of organic content in water can facilitate improved wastewater treatment and better protect the environment. Microbial fuel cells (MFCs) are increasingly recognized as valuable biological monitoring systems, due to their ability to swiftly detect organic indicators such as biological oxygen demand (BOD) and chemical oxygen demand (COD) in water quality. Different types of MFC sensors are used for BOD and COD detection, each with unique features and benefits. This review focuses on different types of MFC sensors used for BOD and COD detection, discussing their benefits and structural optimization, as well as the influencing factors of MFC-based biomonitoring systems. Additionally, the challenges and prospects associated with the development of reliable MFC sensing systems are discussed.

## 1. Introduction

Biochemical oxygen demand (BOD) is a vital parameter for assessing water quality, quantifying the extent to which microorganisms consume dissolved oxygen during the decomposition of organic substances [1]. This parameter offers insights into the pollution level of a water body, caused by organic substances. The impact of high BOD levels on the environment is significant, as dissolved oxygen is essential for the survival of aquatic plants and animals. High BOD levels can cause microorganisms and bacteria to consume the dissolved oxygen, leading to suffocation of other aquatic species. Furthermore, elevated BOD levels increase the content of pollutants, such as nutrients, pathogens, and algae [2,3,4], thereby causing a decline in water quality and disturbing the equilibrium of the ecosystem. Timely monitoring of BOD in real time plays a crucial role in swiftly detecting issues with water quality and implementing suitable actions to tackle them. This kind of monitoring is particularly valuable in wastewater treatment facilities, as it enables the fine-tuning of treatment processes and parameters to optimize efficiency and effectiveness [5]. However, traditional BOD detection methods are often time-consuming and labor-intensive, underscoring the need for the development of rapid and convenient alternative methods for on-site BOD monitoring.

Biosensors have become increasingly favored as analytical instruments for environ-mental surveillance owing to their portable nature and capability of assessing biological impacts [6]. Biosensors can be categorized according to the nature of the biorecognition component employed, including immunosensors [7], DNA biosensors [8,9], cell-based biosensors [10], and biosensors [11]. Enzymatic biosensors, which use immobilized enzymes as biorecognition elements, are more sensitive and specific in detecting BOD [12], but the purification and immobilization of enzymes are costly and time-consuming. Additionally, enzymes have a short shelf life and are highly susceptible to inactivation [13], limiting the usefulness of enzymatic biosensors. On the other hand, BOD sensors that use microbial cells as biorecognition elements are more robust and have a long shelf life, eliminating the need for expensive enzyme purification. Furthermore, microbial cells do not have strict operating conditions, making them more convenient to use [10,14]. As a result, BOD sensors that use microbial cells as biorecognition elements are often considered as a promising alternative to enzymatic biosensors.

Microbial fuel cells (MFCs) find extensive application in environmental monitoring as cell-based biosensors. Karube [15] developed the first MFC-type BOD sensor using immobilized soil flora and a platinum electrode for measuring BOD in wastewater from slaughterhouses, food plants, and alcohol plants. The steady state was achieved within 30–40 min, and the relative error of the results was less than 10% compared to the 5-day biochemical dilution inoculation method. MFC sensors have demonstrated utility in detecting high levels of chemical oxygen demand (COD) and can be adjusted for the analysis of BOD [16,17]. This significantly reduces the time-consuming traditional tests, and makes MFCs an efficient and promising tool for real-time BOD (or COD) monitoring. Conventional MFC sensors are susceptible to temperature and pH, and the manufacturing process is complex, making it difficult to generalize the application [18]. Therefore, current research is focused on improving the stability of MFC sensors by enhancing their design and manufacturing processes, as well as developing different types of MFC sensors for various detection scenarios [19].

This paper provides an overview of the present state and fundamental principles of biosensors, based on MFC. The main focus is on MFC sensors tailored for detecting BOD and COD, along with their intricate structural designs. Additionally, the article briefly delves into the diverse factors influencing MFC sensor performance, such as electron acceptors and electrode materials. The study concludes by exploring potential challenges and future prospects for MFC sensors.

## 2. Working Principles of MFC Sensors

MFC sensors make use of the respiratory metabolism of microorganisms attached to the anode, converting the chemical energy of organic matter present in wastewater into electrical energy. This process creates a connection between the anode and the cathode, resulting in the generation of an electrical current [20,21]. Under certain conditions, the intensity of the electrical output signal, such as current and transferred charge, is proportional to the BOD in the substrate (Figure 1). There are three methods for analyzing electric signals in MFC sensors [5,22].

In the current approach, which relies on the maximum output current generated by microbial oxidation of organic matter per unit anode area, this analytical signal varies in accordance with the concentration of organic matter being detected. This method enables fast, continuous, online BOD detection and does not require high instrumentation. However, it needs to consider the influence of oxygen diffusion, cathode catalytic activity, and other factors, which may lead to unstable or low current signal.

The second method is the coulombic yield(CY) [23]. This method quantifies the total number of electrons (Q) generated by the MFC during the oxidation of organic matter within a specific timeframe. Q is then correlated with the concentration of biodegradable organic matter (BOM) at the anode, demonstrating a linear relationship with BOD. The CY method proves highly efficient in rapidly and accurately determining BOM levels. Yuan Liu [24] applied this method to detect BOM, and achieved promising outcomes. Experimental results revealed that biodegradable organic matter electronic yield (BOM-Q) ranging from 5 to 500 mg/L could be directly obtained within hours to tens of hours, without the need for calibration, with an error of less than 5%. Notably, the ratio of BOM-Q values to BOD_5_ values for both synthetic and real wastewater samples was approximately 1.0. Moreover, the BOM-Q assay maintained its accuracy and precision across various optimized operating conditions. When a large linear concentration range is required, using CY has advantages over current output, because the reaction time remains proportional to the concentration of organic matter even if the reaction rate reaches saturation. The greatest challenge with CY is maintaining constant coulometric efficiency (CE) when BOD concentrations vary. Moreover, there is dynamic competition between bacterial growth within the anode biofilm and electron production and transfer [25], leading to lower CE and thus lower CY values.

The maximum voltage method is a third method that relies on the peak voltage output of the MFC sensors [26], which determines BOD by measuring the maximum power point voltage of the MFC sensor or the voltage when the circuit is open. While this method avoids the influence of external load on sensor performance, its measurement accuracy is relatively low. As an illustration, in the study conducted by W. Logrono [27], organic matter concentrations were monitored using an open circuit voltage, but this approach only yielded two valid BOD values.

Despite the favorable linear relationship between electrical signals and BOD demonstrated by methods like the CY method and the current method, certain limitations have been acknowledged in their application. Consequently, building upon the CY method, the researcher proposed the partial coulomb yield (P-CY) method. In this approach, the point of maximum voltage drop rate is used as the cut-off point for voltage collection [28]. Data collected after the voltage reaches the cut-off point are no longer collected, and the data collected before this point represent the partial CY. Gao Yang [29] utilized this method for the assay, and found that the use of P-CY allowed for higher concentrations of BOM (37.5–375 mg/L) compared to CY, with significantly improved accuracy (R^2^ = 0.999) and relatively shorter reaction times (0.99 ± 0.18~18.08 ± 0.58 h). These results suggest that P-CY can be a more efficient and reliable method for the detection of BOM in wastewater treatment applications. Overall, these novel methods offer valuable alternatives to traditional BOD measurement techniques, with potential for rapid and accurate determination of BOM. Future research may further refine these methods and optimize their use in practical application.

## 3. MFC-BOD Sensor Construction Type

Depending on the device, there are two common types of MFC sensors: dual-chamber MFC (Figure 2A) and single-chamber MFC (Figure 2B). In addition to the MFC sensor research, subsequent researchers have developed new biosensor devices, such as microbial electrolytic cell (MEC) (Figure 2C, D), miniaturized MFC (Figure 2E), multi-stage MFCs (Figure 2F), submersible MFC (Figure 2G), and coupled MFC (Figure 2H). These various types of MFC sensors are used to detect BOD or COD, and numerous studies have shown that MFC sensors have a good detection range and response time (Table 1).

### 3.1. Dual-Chamber MFC Sensor

The dual-chamber MFC sensor is composed of an anode, a cathode, and an ion ex-change membrane, with a diaphragm separating the anode from the cathode to prevent the diffusion of cathodic fluid into the anode compartment (Figure 2A). Despite the complex design, the dual-chamber MFC offers enhanced stability to the electron transfer process, resulting in higher coulombic efficiency [53]. In the vicinity of the anode, microorganisms oxidize organic matter and subsequently transport electrons to the anode using a promoter or mediator (e.g., non-primitive carriers or chemicals like potassium ferricyanide and methylene blue) [54,55], or directly through microbial respiratory enzymes. Nevertheless, chemical mediators are not only costly, but also pose toxicity risks to microorganisms near the electrodes, leading to the death of significant numbers of microorganisms and potential contamination of the ion-exchange membrane. As a result, mediator-less MFC sensors are more practical for broader utilization. For instance, Hsieh [33] designed a dual-chamber MFC sensor without the need for chemical mediators. In the evaluation of BOD concentrations in real wastewater, the MFC sensor exhibited remarkable precision, yielding a deviation of merely 2.5% to 3.6%. It successfully gauged BOD concentrations as high as 240 mg/L. Similarly, Minh Hang Do [26] developed a mediator-less dual-chamber MFC sensor, using a mixed inoculation of cultured microorganisms in a carbon felt anode, which enabled a wider range of BOD detection concentrations, up to 300 mg/L, and demonstrated stable operation for up to 30 days. These studies have confirmed the effectiveness, reproducibility, and high linearity of the mediator-less dual-chamber MFC-BOD sensor, making it suitable for commercial applications. However, incorporating isolation materials in dual-chamber MFCs results in elevated equipment costs and can cause substantial pH variations between the cathode and anode chambers, which can inhibit anode biofilm activity [56,57].

### 3.2. Single-Chamber MFC Sensor

The single-chamber microbial fuel cell (SCMFC) has a relatively simple structure, with both the cathode and anode situated within a single reaction chamber, eliminating the requirement for material separation (Figure 2B). Its main advantage is the avoidance of complicated and tedious operation steps, such as aeration, regeneration, and cathode liquid recovery, which greatly reduces the operation cost. M. Di Lorenzo [34] designed, for the first time, a simple and compact SCMFC as a volatile organic compounds biosensor. The sensor demonstrates a linear response at COD concentrations up to 500 ppm, which increases the dynamic monitoring range by 133% compared to a double-chamber sensor. Moreover, the sensor shows good reproducibility; in an SCMFC sensor without an exchange membrane, one of its advantages is the reduction of the effect of the pH gradient on microorganisms [58]. In the SCMFC with an air cathode, the cathode electrode is positioned in the air, eliminating the necessity for cathode aeration. This configuration has been demonstrated to be effective for monitoring the BOD of wastewater containing suspended and colloidal organic matter, such as corn starch and milk [28]. Furthermore, Ying Wang et al. [31] devised an SCMFC sensor with an activated carbon catalyzed air cathode. Through simulation with synthetic wastewater, they achieved an upper detection limit of 1280 mg/L BOD, demonstrating a significant improvement in the detection range compared to other SCMFC devices.

Despite the advantages of the SCMFC, its oxygen leakage issue presents a challenge, as it slow down metabolism of electro-activated bacteria (EAB) on the electrode surface, resulting in reduced coulombic efficiency and weakened sensitivity of the sensor [30]. To overcome this challenge, one effective solution is to use a membrane electrode assembly (MEA) in the SCMFC. The MEA consists of a proton-conducting polymer film that is sandwiched between two electrodes (the anode and cathode) made of platinum or platinum alloy. These electrodes act as catalysts that accelerate the chemical reactions at each electrode and prevent the diffusion of gases from the air cathode [59,60]. Mia Kima and colleagues utilized a platinum-catalyzed carbon cloth cathode in combination with a cation-exchange membrane and air cathode to successfully construct an MFC-BOD sensor. The incorporation of the MEA resulted in significant improvements, as evidenced by the increase in coulombs from 0.52 C to 4.65 C, and in power from 880 m W/m^3^ to 69,080 m W/m^3^, when the BOD concentration was at 200 mg/L. moreover, it demonstrated good linearity (R^2^ = 0.97) and over 97% repeatability, indicating its high degree of accuracy and reliability [61]. While the low maintenance and easy automation through voltage SCMFC sensors make them highly applicable, one notable drawback is their longer response time compared to dual-chamber MFC sensors and pH gradient [62].

### 3.3. MEC Sensor

MFC sensors incorporating electro-producing bacteria show great promise as devices for detecting organic matter in water. However, in practical applications, these sensors face challenges such as long response times, signal hysteresis, and limitations in both upper and lower detection limits. One way to overcome these limitations is by applying a voltage outside the MFC to create a microbial electrolytic cell (MEC) [63]. This can be achieved using common types of MEC sensors, such as the SCMFC applied voltage (Figure 2C) and the dual-chamber MFC applied voltage (Figure 2D). The use of MEC sensors has been shown to enhance the upper limit of COD detection, as demonstrated by Yuan et al. [38]. By applying a voltage of 1.2 V, they effectively augmented the COD detection range by an additional 130 mg/L within a mere 3 min timeframe, surpassing the initial limit of 75 mg/L. Similarly, Adekunle et al. [37] utilized a single-chamber MEC sensor to measure COD in brewery wastewater in real time. They found a strong correlation (R^2^ = 0.98) between the power output of the MEC and the COD concentration in the anode chamber; the MEC sensor’s upper detection limit was over 500 mg/L.

In addition, Oskar Mo din et al. [35] successfully detected BOD concentrations spanning 32–1280 mg/L (R^2^ = 0.97) by introducing an additional input voltage of 0.2 V and utilizing am MEC sensor throughout a reaction period of 20 h. As the response time decreased, the detection range of the device also diminished, with an observable upper limit of 320 mg/L for measurable BOD concentration values at a response time of 5 h. Wang et al. [36] developed an innovative MEC sensor with an EAB-loaded electrode and a constant external voltage of 0.7 V. This sensor demonstrated the remarkable ability to swiftly detect BOD concentrations ranging from as low as 10 mg/L to as high as 500 mg/L within a short span of five minutes. Moreover, it exhibited the capacity to respond to fluctuations in organic matter within the water column in under one minute. Overall, these studies demonstrate the versatility and potential of MEC biosensors for detecting and measuring various types of organic pollutants in wastewater. However, it is crucial to note that MEC sensors require an external power source, unlike MFC sensing, which is self-powered, and this factor may limit their application and development.

### 3.4. Multi-Stage MFC Sensor

As the concentration of BOD increases, the biofilm volume on the anode can become saturated, leading to a stagnation of current production by bio-redox, which in turn can prevent the detection of MFC sensing at high BOD concentrations [64]. To overcome this limitation of single-stage MFC, multi-stage MFCs (MS-MFCs) have been developed, Through the integration of multiple MFCs, the MS-MFCs system effectively utilizes water from the upper MFC as the inflow for the lower MFC. This design enhancement not only improves the organic matter sensing efficiency, but also enhances the sensitivity and detection range of the sensor (Figure 2F). Wastewater is treated sequentially at each stage, allowing for more complete substrate utilization and higher sensitivity to changes in BOD levels [65]. In the study conducted by Martin et al. [40], they devised a three-stage MS-MFCs system consisting of three MFC connections. This innovative setup effectively extended the detection range by calibrating the sum of the current densities of individual MFCs within the system. As a result, they achieved BOD concentration detections of up to 720 mg/L (R^2^ = 0.97), surpassing the upper detection limit of 360 mg/L attainable with a single MFC.

The MS-MFCs sensor offers the crucial benefit of mitigating the impact of toxic substances on BOD detection. While the microorganisms in the first stage of the MFC are affected by toxicity, which reduces the output current and voltage, the subsequent stages can weaken this effect. This attribute enables the detection of BOD while also identifying the presence of toxic substances. Martin et al. [41] utilized an MS-MFC sensor to differentiate between signal reductions related to changes in BOD concentration and those related to toxic compounds. They started by gradually reducing the BOD from 360 mg/L to 60 mg/L, which resulted in current density reductions of 59%, 82%, and 94% in the three interconnected MFCs. In another set of experiments, they added 4-nitrophenol (150 mg/L) as a toxic compound, which led to a 63%, 66%, and 74% decrease in current density for the MS-MFCs, respectively. The distinct levels of current density reduction allowed for differentiation between the effects caused by the toxic substance and changes in the concentration of organic matter. Similarly, Godain et al. [66] developed a four-stage MFC that demonstrated the feasibility of detecting the biodegradability of toxic compounds. The current intensity of the first MFC decreases gradually, owing to the presence of toxic compounds. However, in the subsequent stages of the MS-MFCs, where most of the previous toxic organics have been degraded, the current intensity only slightly decreases, and then gradually recovers. The latter stage of the MS-MFCs can then be used as a detection stage for toxic compounds.

However, there are also some disadvantages to using MS-MFCs. One of the main challenges is the complexity of the system, which requires more components and maintenance compared to the single MFC. The use of multiple chambers also increases the potential for membrane fouling, and requires careful monitoring to prevent clogging [67]. Furthermore, the higher cost of construction and maintenance may limit the scalability and practicality of MS-MFCs for applications.

### 3.5. Miniaturized MFC Sensor

Conventional MFC-based sensors typically have relatively long response times, often several hours (Table 1); starved microorganisms at low concentrations of organics can rapidly degrade organics to shorten the response time, which does not reflect real-time wastewater conditions. The volume of the MFC chamber is critical to the response time (Figure 2E). Studies have shown that reducing the reactor volume from 50 mL to 12.6 mL resulted in a 77.3–83.3% reduction in response time [34]. In another study, using a volume of 2 mL of SCMFC, 3–164 ppm of COD was detected with a response time of only 2.8 min [45]. The miniaturized MFC offers a high specific surface area, enabling the optimal utilization of organic matter and resulting in higher energy densities. This advantage arises from the small size of the reactor chamber and low internal resistance, which contribute to enhanced performance and efficiency [68,69]. These advantages make the miniaturized MFC unit faster in response time.

To achieve faster response times and a wider detection range, Xiao et al. [43] utilized xurographic fabrication technology [70] to produce a miniaturized MFC sensor with a reaction volume of just 1.8 mL. The researchers conducted tests on the sensor using acetic acid as the BOD source, and they observed its respond to BOD concentrations spanning from 20 to 490 mg/L, with a response time of only 1.1 min. Despite offering significant advantages, miniaturized MFC sensors also have some drawbacks. For instance, their small size can make it challenging to maintain stable microbial communities, and the low power output of miniaturized MFCs may limit their sensitivity and accuracy in detecting low concentrations of BOD [46,71].

### 3.6. Submersible MFC Sensor

Submersible MFC sensors have garnered significant attention, owing to their capability of providing real-time, in situ monitoring of BOD in bodies of water. When deployed in lakes, rivers, and sewage anaerobic ponds, these sensors can be directly placed at the bottom, taking advantage of the favorable anaerobic environment for the MFC. The cathode can be positioned on the water’s surface or in the middle, to facilitate aeration. Notably, the submersible MFC eliminates the need to collect water samples and conduct separate testing, as was the practice with previous MFC sensors and traditional BOD detecting methods [72]. Additionally, submersible MFC sensors can be employed for long-term monitoring, providing continuous data that can detect changes in water quality over time. The submersible MFC was designed to be simpler by utilizing an air cathode and immersing the anode electrode in an anaerobic reactor, eliminating the need for an anode chamber (Figure 2G). An alternative type of submersible MFC utilized as a BOD sensor involves immersing the anode electrode directly into the anaerobic wastewater while continuously flushing the cathode chamber with air. Moreover, a pressurized membrane is affixed to the cathode side, effectively minimizing the internal resistance of the cell. With this method, the MFC sensor is capable of detecting 78 ± 8 mg/L of BOD within a period of 10 h [48]. Another study applied the submersible MFC to monitor the response of activated sludge (AS) reactors to shocks from high-concentration pollutants. The results showed that under a high-concentration COD shock for 1.5 h, the unit gave a warning signal when the output voltage decreased by 0.04 V. Remarkably, this warning signal was observed 4.5 h before the water quality began to deteriorate. This established a new online monitoring and alarm system with self-diagnostic functions for the AS process [73]. Despite the numerous benefits of submersible MFC sensors, they are also faced with several challenges. One of the most significant issues is their susceptibility to electrode fouling, which occurs due to exposure of the anode to sewage or sludge, leading to a decline in sensor performance over time. Additionally, maintaining a consistent supply of oxygen to the cathode can be challenging for submersible MFC sensors when operating in anaerobic wastewater.

### 3.7. Coupled MFC Sensor

In recent years, an emerging integrated platform has garnered interest by combining constructed wetlands with MFC technology (CW-MFC) [74,75]. Researchers have successfully used this approach to treat various types of polluted wastewater [76,77,78]. Coupling the MFC with vertical flow constructed wetlands (VFCWs) is a more efficient approach, because it maximizes the use of redox gradients in constructed wetlands and creates an environment conducive to EAB growth [79]. In addition, introducing granular electrodes, such as activated carbon (AC) and graphite gravel (GG) [80], can enhance the conductivity of EAB, resulting in improved system performance and stability. By utilizing MFC sensors, the VFCW-MFC system can monitor specific pollutants, such as COD and BOD (Figure 2H), in real time, and make necessary adjustments for improved removal efficiency [81]. Xu et al. [51] used VFCW-MFC for COD monitoring, in which dewatered alum sludge was used as the substrate and the cathode consisted of stainless steel mesh combined with AC, while GG was used as the anode. The MFC electrodes were positioned in the aerobic and anaerobic sections of the VFCW. The study found a strong linear correlation was observed between the output voltage and COD concentration, spanning the COD concentration range of 0 to 1000 mg/L. Due to its size, the placement of electrodes in VFCW-MFC has a significant impact on the current density and detection range [82]. EAB are typically more suited for anaerobic and anoxic environments [83], so the bottom or middle of the VFCW is used to bury the anode, where there is a low-oxygen redox potential. The upper surface of the VFCW can house the cathode, where it can use oxygen in the air as an electron acceptor [84]. The spacing between the electrodes affects the output signal of the sensor, and large electrode spacing leads to a decrease in current density, which can be improved by increasing the anode volume. To determine the best structure for VFCW-MFC, researchers proposed using the coefficient S (S = L_A_/L_S_, where L_A_ is the anode size and L_S_ is the electrode spacing) [50]. Studies showed that the S = 0.75 structure had better sensitivity (160.05 C/mg, R^2^ = 0.9805) when COD concentrations were 400–1000 mg/L. However, the response time of S = 1.5 was shorter (2.2–17.8 h, R^2^ = 0.9996) when the COD concentration was lower than 400 mg/L. In general, a larger S leads to a shorter response time, while a smaller S has better sensitivity. It is worth noting that while CW-MFC can provide an initial indication of water quality, its accuracy may decrease over time, and it may not be suitable for precise monitoring of COD [49].

The use of up-flow anaerobic sludge blanket (UASB) reactors in wastewater treatment has become widespread, due to their high organic removal rate and ability to adapt to changes in temperature and pH [85,86]. The effluent COD concentration is often used to assess whether the effluent in a UASB reactor is being treated to standard, but traditional monitoring methods are not able to provide real-time online monitoring. A new UASB-MFC coupling system has been developed that employs the MFC as a sensor to monitor the COD effluent from the UASB [52]. In this system, the anoxic environment inside the UASB reactor can be used to house the anode, and the cathode can be exposed to the air. The voltage response of the system was found to increase linearly over 12 h as COD levels rose from 500 to 3000 mg/L (R^2^ = 0.9368), but the voltage declined linearly (from 0.32 ± 0.01 to 0.17 ± 0.01 V) as COD rose from 3000 to 5000 mg/L (R^2^ = 0.9326). This suggests that the UASB-MFC sensor COD measurements may be influenced when readings are outside of a specific range. Jia et al. [87] conducted a study in which they developed a dual UASB-MFC sensor system with sensors located in both the suspension and sludge layers. The study found that MFC located in the suspension layer was more effective in monitoring wastewater COD (R^2^ = 0.987), while in the sludge layer it was better suited for the monitoring of total volatile fatty acids with high accuracy (R^2^ = 0.997). This demonstrates the versatility and potential of using MFCs as sensors in UASB-MFC coupling systems for monitoring multiple parameters in real time.

### 3.8. Comprehensive Evaluation of MFC Sensor

Various types of MFC sensors have demonstrated commendable detection capabilities, but they each come with their own set of limitations (Table 2). Sensors with intricate structures and larger volumes tend to exhibit improved stability, due to their larger reaction chambers, which can better accommodate changes in organic loading. On the other hand, miniature MFC sensors boast rapid response times but may lack stability, and mainly rely on synthetic wastewater for detection, necessitating further refinement before practical real-world applications. Complex structures may lead to increased costs, as seen in dual-chamber MFC sensors that employ ion-exchange membranes, which are prone to contamination and require frequent replacement, making them cost prohibitive [88]. Meanwhile, SCMFC sensors not using an exchange membrane face challenges related to pH gradients. MEC sensors use external voltage to reduce response time and mitigate rapid membrane contamination, yet the reliance on an external power supply hinders their widespread adoption. Although MS-MFCs expand the detection range, they also introduce additional challenges. Depositional MFCs and coupled MFCs possess heterogeneous structures and can be effective in specific environments. However, coupled MFCs are limited to specific systems and only applicable to single scenarios. In conclusion, various MFC types can be chosen for different environments and requirements, and specialized MFCs can be developed for unique environmental settings.

## 4. Performance Parameters and Optimization Strategies

### 4.1. Environmental Parameters

Environmental Parameters that affect the performance of MFC sensors for organic monitoring include operating temperature, fuel type, pH, and external resistance. According to Ma, Y Ma et al. [18], there is a clear linear range between the output voltage of the biosensor and the BOD values at different operating temperatures: 50 mg/L at 15 ± 1 °C (R2 = 0.9855), 70 mg/L at 25 ± 1 °C (R^2^ = 0.9917), and 80 mg/L at 35 ± 1 °C (R^2^ = 0.9928). Peixoto et al. [48] evaluated the performance of the MFC sensor under different pH conditions using domestic wastewater with a BOD of 144 ± 9 mg /L. Notably, the biosensor exhibits an exceptional maximum current density of 288 mA/m^2^ at pH 7.0. However, a decline in performance is evident at both acidic and alkaline pH levels, with the minimum current densities recorded at 186 mA/m^2^ (pH 6 ± 0.1) and 184 mA/m^2^ (pH 8 ± 0.1), respectively.

The calibration range is also notably influenced by the external resistance (R_e_). A com-parison between R_e_ = 43.2 Ω and 305 Ω reveals a 60 mg/L decrease in the linear calibration range [90], which ranges from 15 to 180 mg/L BOD. Subsequently, at elevated R_e_ values (953 Ω and 5100 Ω), the calibration range is further constrained to 15–150 mg/L and 15–30 mg/L, respectively. The fluid dynamics within the anodic chamber significantly affect the sensitivity of the MFC sensor [5]. To enhance the performance, Yue Yi et al. [91] carried out optimization of the anodic structure, leading to a notable 14.1% reduction in the dead zone proportion and an impressive 334.6% increase in the anode surface velocity. After the optimization, the sensitivity of the MFC sensor exhibited a substantial 52.3% increase.

### 4.2. Diffusion of Oxygen and Electron Acceptors

Another challenge is the diffusion of oxide substances from the cathode to the anode, like nitrate and oxygen. These substances rob electrons from the biofilm, lowering the CE value and ultimately reducing the electrical signal from the sensor [92]. The elimination of this effect can be achieved by adding terminal oxidases along with de-nitrification inhibitors like sodium azide and cyanides [93]. However, these chemicals are harmful to mammalian cells, because they interrupt extracellular electron transfer. Low permeability membranes have been developed to reduce losses due to oxygen diffusion. The researchers conducted a study where they prepared an SCMFC to monitor BOD in wastewater, employing sulfonated poly ether ether ketone (SPEEK) membranes [30]. Compared to the Nafion-based MFC, the new membranes had lower oxygen permeability. The MFC with SPEEK membranes showed a 62.5% increase in the upper measurement limit and more than 76% increase in output current.

The conventional MFC sensors utilize dissolved oxygen as the electron acceptor, which is capable of measuring BOD within the range of 0.1 to 200 mg/L [1], and with the upper detection limit usually below 200 mg/L, with a detection time of about 20 h [48]. To enhance the performance of the MFC sensor, a comparison was made between dissolved oxygen and KMnO_4_ as cathodic electron acceptors for wastewater BOD detection [94]. The biosensor with KMnO_4_ at 10 mmol/L exhibited outstanding performance compared to dissolved oxygen, including an expanded upper limit of BOD detection to 500 mg/L, a 50% shorter response time for 100 mg/L BOD in artificial wastewater, and reduced relative error in BOD detection, to below 10%. Additionally, another study established a mediated BOD biosensor, employing ferricyanide as an electron acceptor, featuring a linear range of 4 to 60 mg/L and a BOD limit detection of 1.8 mg/L [95].

In MFC sensors where organic matter coexists with nitrate, the microorganisms in the system are more inclined to utilize nitrate as an electron acceptor, which leads to a reduction in the amount of organic matter available to the electricity-generating bacteria, and ultimately a weakening of the sensor electrical signal output [96]. A recent study observed that standard BOD solutions with BOD levels ranging from 20 to 500 mg/L were employed, along with different levels of nitrate (0–50 mg/L ${NO}_{3}^{-}$-N). The results indicated that the biosensor was unable to accurately measure the BOD concentration of samples containing nitrate. Graphical and mathematical methods were subsequently proposed for adjustment, and the corrected BOD concentration was able to precisely reflect the organic matter levels in the nitrate-containing sample. In comparison to employing denitrification inhibitors like sodium azide to enhance the performance of MFC sensors for BOD detection in nitrate-containing wastewater [93], this method is more practical and promising, due to its efficiency and ease of implementation without the use of toxic chemicals.

### 4.3. Microbial Inoculation

Research has found that microbial populations on electrodes significantly influence the electrical response of MFC. For example, the microbial source used for inoculation during the startup phase of MFCs has a significant impact on subsequent performance [97]. Yue Yi et al. [98] systematically compared the sensitivity of MFC sensors constructed with two different inoculums. The results showed that the mixed-culture MFC exhibited a sensitivity 3.86 times greater than that of the *Shewanella loihica PV-4*-cultured MFC. The anode of the MFC inoculated with mixed strains will have a higher biofilm viability and biomass density, which will increase the sensitivity of organic matter detection [96]. EAB in MFC sensors serves as a crucial component, playing a significant role in BOD detection, but improper maintenance can degrade the detection performance. Fei Guo et al. [99] set the EAB within the MFC sensor to sleep, to allow maintenance of the MFC sensor during downtime. Reactivation of the sensor after a month of dormancy restored the voltage in a short period of time and improved the accuracy in the 200–500 mg/L BOD range.

### 4.4. Response Time

The response time of an MFC sensor is defined as the time required to achieve a new steady state after BOD variation [34]. Typically, a longer response time was observed with a higher BOD [48]. Lorenzo et al. [100] improved the performance of the BOD sensor with an anode designed with a bed packed with graphite granules, and the response time was reduced by about 65% compared to the MFC sensor using a carbon cloth anode when the particle layer thickness was 1 cm and the external resistance was 500 Ω. A reduction in response time can also be achieved by increasing the solution flow rate and reducing the volume within the anode chamber. Increasing the influent flow rate from 15.6 mL/h to 43.8 mL/h reduced the response time by 161 min [30], and reducing the total volume of the anode chamber by 75% reduced the response time of the sensor to 40 min [34]. For the miniaturized MFC sensor, low (<10 μL/min) or high (>70 μL/min) flow rates of the sample solution to be measured result in uneven substrate distribution and, therefore, weakened growth and metabolism of microorganisms at the edge of the biofilm [101]. In this study, the effective detection range of sodium acetate was 0.1~0.7 g/L (50~490 mg/L of BOD) at the optimal flow rate (50 μL/min).

### 4.5. Electrode Materials

The electrode material plays a critical role in MFCs, significantly influencing the sensor detection capability. Specifically, the nature and structure of the anode material have a profound impact on MFC sensor performance. The anode serves as a site for microbial loading, and requires good biocompatibility and conductivity [102]. Carbon-based anodes are widely preferred in MFC sensors, due to their high specific surface area, porosity, excellent electrical conductivity, and corrosion resistance (Table 1). Among carbon-based materials, the specific surface area (SSA) becomes a crucial factor affecting their performance [103]. In low concentration (25~400 mg/L) BOD assays, CF outperformed CC as an anode substrate material. CF features a loosely packed carbon fiber structure with a higher SSA, resulting in higher BOD response values and more stable coulombic efficiency [23]. Furthermore, carbon nanotubes (CNT), renowned for their high SSA and mechanical stability, are commonly utilized for anode modification in MFCs [104]. CNT treatment could increase the proportion and current generation of electroactive microorganisms, and also improve the BOD linear range, compared to untreated sensors (from 49~321 mg/L to 129~492 mg/L) [44].

Cathode reduction is the bottleneck reaction in microbial fuel cell systems [105,106]. Therefore, the kinetic process of the reaction can be accelerated by adding catalysts to lower the cathodic activation overpotential. Platinum (Pt) catalysts are heavily used in MFC applications; however, they have been abandoned in recent years because of their high cost and limited durability [107]. Khar et al. [108] used the lower-cost β-MnO_2_ (instead of Pt) as the oxygen reduction catalyst in the MFC BOD sensor. The real domestic wastewater system with β-MnO_2_ as catalyst showed good stability (over 1.5 years of operation) and the voltage remained within a good linear relationship (R^2^ = 0.93) with BOD_5_ values (33~160 mg/L). Another study utilized AC as the cathode catalyst: CY was linearly correlated (R^2^ = 0.9965) with the sodium acetate BOD concentration within 80~1280 mg/L 80~1280 mg/L when the external resistance was 1 KΩ [31].

### 4.6. Substrate Effects

The type and content of the substrate, which serves as the energy source for bacteria on the MFC electrode, significantly influence the strength of the signal output to the sensor [109]. A study assessed how different substrates for electricity generation in the MFC affected the calibration curves that related BOD concentration to MFC output signals [110], and results indicated that monosaccharides and disaccharides were better substrates than amino acids, organic acids, or alcohols. Furthermore, different concentrations of organic matter affect the colony structure within the MFC sensor. When researchers injected varying concentrations of glucose and glutamic acid mixed solution (GGA) into the sensor, they found that within the range of GGA concentrations from 37.5 mg/L to 375 mg/L, the anodic microorganisms were mainly the fermentative bacterium *Raoultella*. Its main function was the degradation of GGA into smaller organic molecules, which other microorganisms then utilized for electricity production. On the other hand, when the substrate contained formaldehyde, the microbial community of the MFC sensor changed as the concentration of formaldehyde increased. The abundance of *Geobacter*, the most dominant EAB, decreased from 59.69% to 44.38% [111]. This change resulted in a decrease in the output voltage of the sensor, indicating that excessive concentrations of toxic organics can harm the bacteria within the MFC, leading to a potential loss of detection capability.

## 5. Challenges and Perspective

While MFC sensors show great potential in organic monitoring, there are a number of challenges that need to be faced before they become a widely used technology. One of the challenges associated with developing an MFC BOD sensor is the selection of appropriate bacteria. The bacteria used in the sensor must be capable of efficiently oxidizing organic compounds present in the sample and generating enough electrons to produce a measurable electrical signal. However, with prolonged operation, the colony structure on the electrodes can change with the environment, which may negatively impact the detection limits, selectivity, and stability of the biosensor. To overcome this challenge, the construction of genetically engineered microorganisms is a promising strategy for enhancing biosensor performance [112]. Furthermore, identifying metabolic pathway genes linked to sensor-related EAB has the potential to enhance the detection capabilities of MFC sensors and broaden their detection capabilities to encompass other substances beyond the current scope.

Another major challenges in real-time monitoring of wastewater in practical applications is signal interference resulting from the combined influence of BOD and toxicants [113,114]. The microbial activity is influenced by toxicants, leading to false signals from MFC sensors [115]. Although MS-MFCs sensors can recognize toxic substances and organics, this approach inevitably faces some issues. Since the generated electrons depend on the BOD concentration, specific toxic compounds cannot be identified in the presence of insufficient BOD concentrations. Therefore, low BOD concentrations result in a low output signal strength, which makes the detection of toxic substances difficult. MFC sensors based on biocathode sensing elements can be used and detection algorithms can be developed for the quantification of toxic shocks [116,117]. This approach may help prevent signal interference and improve the accuracy of real-time monitoring within the monitoring of organics. Additionally, VFCW-MFC sensor systems have electrical signals that are less affected by specific chemical disturbances such as Cu^2+^ and herbicides [81]. However, these sensors suffer from disadvantages such as inaccurate accuracy and long response time. Optimizing the structure of VFCW-MFC sensors and selecting suitable materials for fillers to overcome these limitations will be the focus of research. Miniature MFC sensors are the most promising for widespread use because of their short response time, inexpensiveness, and ease of manufacturing, but they face the challenge of low current density; this can be addressed by using more closely spaced electrodes, increasing the conductivity of the solution, and preventing biological contamination [118]. On the other hand, submersible MFC sensors offer strong application capabilities for real-time detection, but they suffer from electrode corrosion. To mitigate this issue, appropriate electrode materials can be selected [107], substrate concentration and operating conditions can be controlled, and cathodic protection techniques can be utilized.

The process of converting raw data from MFC online sensors into useful water quality parameters is a crucial step in ensuring accurate and reliable readings. However, this task can be challenging, as MFC sensors utilize complex detection algorithms that are not always perfect or uniform across all sensors [119]. As such, we believe that the incorporation of artificial intelligence (AI) may hold the key to improving the accuracy and versatility of MFC sensors. By integrating AI with MFC sensors, the powerful data processing capabilities of AI are used to enhance signal conversion and improve the accuracy of the resulting water quality parameters. This cutting-edge approach has the potential to completely change traditional detection methods by expanding the range of possible applications for MFC sensors and enabling more precise and efficient analysis of water samples.

## 6. Conclusions

In conclusion, MFC sensors have proven to be a promising and versatile technology for monitoring water quality. As discussed in this paper, different types of MFC sensors exist, each with their unique advantages and limitations. Several factors, including pH, temperature, and substrate concentration, can influence the performance of these biosensors. However, with appropriate operational optimization and experimental design, MFC sensors can provide reliable and accurate measurements of water quality parameters. Further research in this field is necessary to advance the development and application of MFC sensors for real-world monitoring scenarios. Ultimately, the integration of MFC-based biosensors into existing water quality monitoring systems could significantly improve the efficiency and accuracy of water detection to better manage and preserve our vital water resources.

## Figures and Tables

**Figure 1 bioengineering-10-00886-f001:**
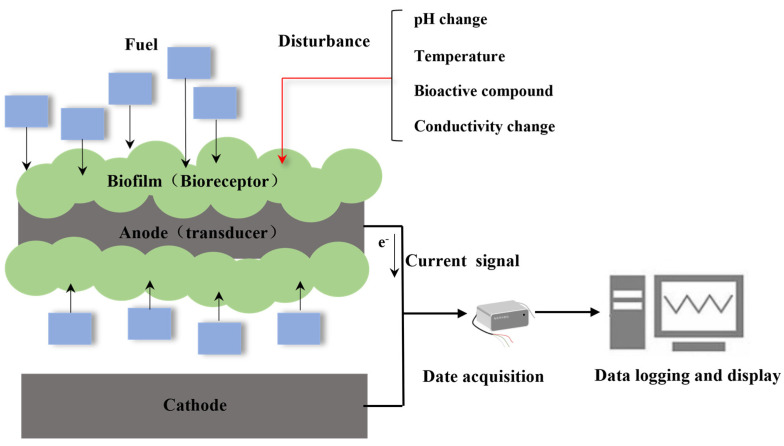
MFC sensor working principle.

**Figure 2 bioengineering-10-00886-f002:**
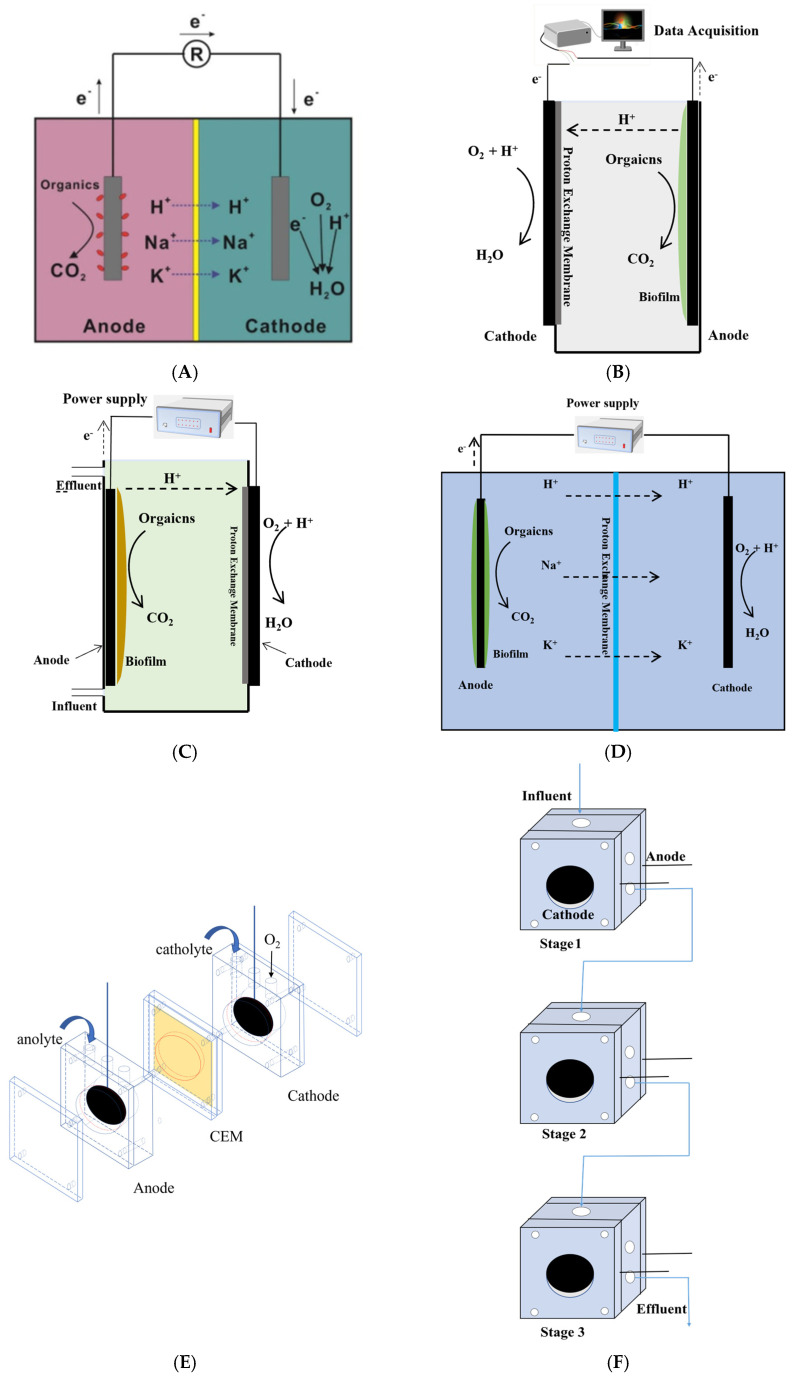
MFC sensor for different devices: (**A**) dual-chamber MFC, (**B**) single-chamber MFC, (**C**) single-chamber MEC, (**D**) double-chamber MEC, (**E**) miniaturized MFC, (**F**) multi-stage MFCs, (**G**) submersible MFC, (**H**) VFCW-MFC.

**Table 1 bioengineering-10-00886-t001:** Summary of different types of MFC sensors for organic detection.

Type	Items	Reactor	Anode	Cathode	Detection Range	Response Time	Signal Acquisition	Ref.
MFC	BOD	Single	GB	CG	59–660 mg/L	0.5–4 day	CY	[28]
		Single	CC	CC/Pt	0–650 mg/L	80 min	current	[30]
		Single	CF	AC	80–1280 mg/L	50 h	CY	[31]
		Double	CF	CFB	0–300 mg/L	N/A	voltage	[26]
		Double	AG	AG	0.34–9.6 mg/L	30–130 min	voltage	[32]
		Double	CC	CC	37.5–375 mg/L	0.99–18.08 h	P-CY	[29]
		Double	CF	Pt/C	5–500 mg/L	5–50 h	CY	[24]
		Double	GF	GF	8–240 mg/L	6–12 h	voltage	[33]
	COD	Single	CC	CC/Pt	0–500 ppm	4.5 h	current	[34]
MEC	BOD	Single	GR	CP	32–1280 mg/L	20 h	CY	[35]
		Double	CF	SSM	10–500 mg/L	5 min	current	[36]
	COD	Single	CF	CC/Ni	100–700 mg/L	48 h	current	[37]
		Single	GFB	SSM	0–130 mg/L	3 min	current	[38]
multi-stag	BOD	Single	CF	AC/Pt	0–149.7 mg/L	61 min	voltage	[39]
		Single	CC	Pt/C	0–720 mg/L	2.3 h	current	[40]
		Single	CC	Pt-GDE	60–360 mg/L	2.3h	current	[41]
Miniature	BOD	Single	CC	CC	9.8–19,600 ppm	37 ± 2 min	current	[42]
		Double	CC	Pt film	20–490 mg/L	1.1 min	current	[43]
		Double	CC/SWCNT	Pt film	129–492 mg/L	N/A	current	[44]
	COD	Single	CC	CC	3–164 ppm	2.8 min	current	[45]
		Double	CC	Pt film	20–400 mg/L	N/A	current	[46]
Submersible	BOD	N/A	CP	CP	0–250 mg/L	3.1 h	current	[47]
		Double	CP	CP	0–78 ± 8 mg/L	10 h	current	[48]
coupled	COD	CW	GR	GG	0–200 mg/L	20 h	voltage	[49]
		CW	AC	GF	50–400 mg/L	2.2–17.8 h	CY	[50]
		CW	SSM/AC	SSM/GC	0–1000 mg/L	12 h	voltage	[51]
		UASB	CF	CC	500–3000 mg/L	12 h	voltage	[52]

CC: carbon cloth; GG: graphite gravel; CF: carbon felt; GB: graphite brushes; CG: carbon granules; CP: carbon paper; GFB: graphite fiber brush; CR: carbon rod; AC: activated carbon; GF: graphite felt; CFB: carbon fiber brush; GR: graphite rod; Pt: platinum; SSM: stainless-steel mesh; N/A: not available; AG: activated graphite; Pt—GDE: Pt gas diffusion electrode; SWCNT: single-walled carbon nano tube; MBCC: manganese-based catalytic carbon; UASB: up-flow anaerobic sludge blanket. CY: coulombic yield; P-CY: partial coulomb yield.

**Table 2 bioengineering-10-00886-t002:** Comprehensive comparison of different MFC sensors.

Type	Advantage	Disadvantage	Cost	Wastewater Samples	Ref.
Dual-chamber MFC	Higher coulombic efficiency, stable structure.	Complex design, cumbersome operation.	average cost	SW, DS, IW	[24,29,32,33]
Single-chamber MFC	Easy to manufacture, simple operation.	PH gradient, low power output,	low cost	SW, DS	[28,30,31]
MEC	Short response time.	requires external power supply.	high cost	SW, DS, IW	[36,37,38]
Multi-stage MFCs	Wide detection range, high resistance to toxic interference.	Manufacturing difficulties, cumbersome operation.	high cost	SW, IW	[39,40,41]
Miniaturized MFC	Short response time, easy to manufacture.	Low power output, unstable detection.	low cost	SW	[42,43,45]
Submersible MFC	Able to monitor on site.	Manufacturing difficulties, cumbersome operation.	high cost	DS, IW	[47,48,73]
Coupled MFC	Real-time monitoring.	Single scope of application, long response time.	high cost	SW, DS	[49,50,51,52,89]

SW: synthetic wastewater, DS: domestic sewage, IW: industrial wastewater.

## Data Availability

Not applicable.

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
