# Peer review of "Microbial Fuel Cell-Based Organic Matter Sensors: Principles, Structures and Applications"

_bioengineering, 2023, doi:10.3390/bioengineering10080886_

Round 1
Reviewer 1 Report
There are many review articles on the same topic published recently. The present paper has an average quality among many others. The paper is not of the high significance scientifically, but it is meaningful as an educational material for students working in this field. Overall, the paper is publishable in the present version (minor editing of the English is recommended) as a potentially readable educational material.
Minor editing of the English is recommended.
Author Response
Dear Reviewers:
Thank you for dedicating your time to review this manuscript. I truly value all your insightful feedback and recommendations! Please find my itemized responses below, along with the revised content in the resubmitted documents.
Point 1: There are many review articles on the same topic published recently. The present paper has an average quality among many others. The paper is not of the high significance scientifically, but it is meaningful as an educational material for students working in this field. Overall, the paper is publishable in the present version (minor editing of the English is recommended) as a potentially readable educational material.
Response 1: thank you for your positive feedback on the article. We have carefully addressed the English expressions that needed improvement (highlighted in red in the manuscript). We sincerely appreciate the valuable time and effort the reviewers have devoted to evaluating our work.
Reviewer 2 Report
Please fix typographical and punctuation errors in the whole manuscript. A thorough proofreading is required.
Please define abbreviations first before referring to them, e.g., Coulombic yield (CY), BOD-I, BOD-Q.
Figure 1 needs improvement as there is no cathode in the illustration.
The discussion and elaboration of both methods compared to traditional methods in lines 105-130 need improvement.
In Table 1, the authors need to mention which of the three stated methods was used for analyzing electric signals in MFCs listed in the table.
Did the authors obtain copyright permission for the reused figures in Figure 2, or did they only provide a citation?
In lines 306-308, "One of the main advantages of these sensors is their ability to offer real-time monitoring without the need for sample collection and transportation to a laboratory for analysis." Please specify the advantage compared to what?
The authors need to compare the applications, pros, and cons of the various MFC biosensors stated in section 3 (3.1-3.7) versus each other, which is lacking in this paper.
One thing that is lacking in this paper is a cost comparison and evaluation which needs to be added.
Please fix typographical and punctuation errors (there are so many) in the whole manuscript. A thorough proofreading is required.
Reviewer 3 Report
Organic matter in water bodies has a huge impact on the environment, and real-time monitoring of organic matter levels in water bodies is an important process for understanding water quality and ensuring environmental safety. Microbial fuel cells (MFCs) have become increasingly popular as biomonitoring systems because they can monitor organic matter such as biological oxygen demand (BOD) and Chemical Oxygen Demand (COD)in water quality in real-time without external power sources. Different types of MFC sensors are used for BOD and COD detection, each with unique features and benefits. This review focuses on different types of MFC sensors used for BOD and COD detection, discussing their benefits and structural optimization, as well as influencing factors of MFC-based biomonitoring systems. Additionally, the challenges and prospects associated with the development of reliable MFC sensing systems are discussed.
In general, the manuscript is written well, and the authors have covered almost all aspects of the functioning of MFCs. However, I would suggest the addition of the following new sections.
A full section on the nature of electrodes should be mentioned, the nature of electrode materials is a compulsory aspect of MFCs.
The effect of resistance and other factors is also desired.
The challenge of microbial inhibition by substrates being treated should also be added.
Author Response
Dear Reviewers:
Thank you for dedicating your time to review this manuscript. I truly value all your insightful feedback and recommendations! Please find my itemized responses below, along with the revised content in the resubmitted documents.
Point 1: In general, the manuscript is written well, and the authors have covered almost all aspects of the functioning of MFCs. However, I would suggest the addition of the following new sections.
A full section on the nature of electrodes should be mentioned, the nature of electrode materials is a compulsory aspect of MFCs.
Response 1: Thank you for recognising our manuscript, the description of the nature of the electrodes has been added in Section 3.7. The modifications are as follows:
“4.5. Electrode Materials
The electrode material plays a critical role in MFCs, significantly influencing the sensor detection capability. Specifically, the nature and structure of the anode material have a profound impact on MFC sensor performance. The anode serves as a site for microbial loading and requires good biocompatibility and conductivity[106]. Carbon-based anodes are widely preferred in MFC sensors due to their high specific surface area, porosity, excellent electrical conductivity, and corrosion resistance (Table 1). Among carbon-based materials, the specific surface area(SSA) becomes a crucial factor affecting their performance[107].In low concentration (25 ~ 400 mg/L) BOD assays, CF outperformed CC as an anode substrate material. CF features a loosely packed carbon fiber structure with a higher SSA, resulting in higher BOD response values and more stable coulombic efficiency[23]. Furthermore, carbon nanotubes (CNT), renowned for their high SSA and mechanical stability, are commonly utilized for anode modification in MFCs[108]. When CNT treatment could increase the proportion and current generation of electroactive microorganisms, and also improve the BOD linear range compared to untreated sensors (from 49 ~ 321 mg/L to 129 ~ 492 mg/L)[44].
Cathode reduction is the bottleneck reaction in microbial fuel cell systems[109, 110]. Therefore, the kinetic process of the reaction can be accelerated by adding catalysts to lower the cathodic activation overpotential. Platinum (Pt) catalysts are heavily used in MFC applications, however, they have been abandoned in recent years because their high cost and limited durability[111]. Khar et al.[112] used the lower cost β-MnO2 (instead of Pt) as the oxygen reduction catalyst in the MFC BOD sensor. The real domestic wastewater system with β-MnO2 as catalyst showed good stability (over 1.5 years of operation) and the voltage remained well linear relationship (R2 = 0.93) with BOD5 values (33 ~ 160 mg/L). Another study utilized AC as the cathode catalyst, CY was linearly correlated (R2 =0.9965) with the sodium acetate BOD concentration within 80 ~ 1280 mg/L 80 ~ 1280 mg/L when the external resistance was 1 KΩ[31]. “
Point 2: The effect of resistance and other factors is also desired.
Response 2: Regarding the resistance and other factors (e.g., hydraulic conditions), we have provided an additional description, in lines 408-418 of the manuscript, as follows:
“The calibration range is also notably influenced by the external resistance (Re). A comparison between Re = 43.2Ω and 305Ω reveals a 60 mg/L decrease in the linear calibration range [92], which ranges from 15–180 mg/L BOD. Subsequently, at elevated Re values (953Ω and 5100Ω), the calibration range is further constrained to 15–150 mg/L and 15–30 mg/L, respectively. The fluid dynamics within the anodic chamber significantly affect the sensitivity of the MFC sensor [5]. To enhance the performance, Yue Yi et al [93] carried out optimization of the anodic structure, leading to a notable 14.1% reduction in the dead zone proportion and an impressive 334.6% increase in the anode surface velocity. After the optimization, the sensitivity of the MFC sensor exhibited a substantial 52.3% increase.”
Point 3: The challenge of microbial inhibition by substrates being treated should also be added.
Response 3: For the effects of substrate on microorganisms, we have added a section on "4.6. Substrate effects", which reads as follows:
“4.6. Substrate effects
The type and content of the substrate, which serves as the energy source for bacteria on the MFC electrode, significantly influence the strength of the signal output to the sensor[113]. A study assessed how different substrates for electricity generation in the MFC affected the calibration curves that related BOD concentration to MFC output signals [114], and results indicate that monosaccharides and disaccharides were better substrates than amino acids, organic acids, or alcohols. Furthermore, different concentrations of organic matter affect the colony structure within the MFC sensor. When researchers injected varying concentrations of glucose and glutamic acid mixed solution (GGA) into the sensor, they found that within the range of GGA concentrations from 37.5 mg/L to 375 mg/L, the anodic microorganisms were mainly the fermentative bacterium Raoultella. Its main function was the degradation of GGA into smaller organic molecules, which other microor-ganisms then utilized for electricity production. On the other hand, when the substrate contained formaldehyde, the microbial community of the MFC sensor changed as the concentration of formaldehyde increased. The abundance of Geobacter, the most dominant EAB, decreased from 59.69% to 44.38%[115]. This change resulted in a decrease in the output voltage of the sensor, indicating that excessive concentrations of toxic organics can harm the bacteria within the MFC, leading to a potential loss of detection capability.”
Round 2
Reviewer 3 Report
No further comments
Almost acceptable